# SYMMETRIC MACHINE THEORY OF MIND

## ABSTRACT

Theory of mind (ToM), the ability to understand others' thoughts and desires, is a cornerstone of human intelligence. Because of this, a number of previous works have attempted to measure the ability of machines to develop a theory of mind, with one agent attempting to understand anothers' internal "mental state". However, ToM agents are often tested as passive observers or in tasks with specific predefined roles, such as speaker-listener scenarios. In this work, we propose to model machine theory of mind in a more flexible and *symmetric* scenario; a multi-agent environment SymmToM where all agents can speak, listen, see other agents, and move freely through a grid world. An effective strategy to solve SymmToM requires developing theory of mind to maximize each agent's rewards. We show that multi-agent deep reinforcement learning models that model the mental states of other agents achieve significant performance improvements over agents with no such ToM model. At the same time, our best agents fail to achieve performance comparable to agents with access to the gold-standard mental state of other agents, demonstrating that the modeling of theory of mind in multi-agent scenarios is very much an open challenge.

## 1 INTRODUCTION

Human communication is shaped by the desire to efficiently cooperate and achieve communicative goals (Tomasello, 2009). Children learn from a young age that the others they interact with have independent mental states, and therefore communicating is necessary to obtain information from or shape the intentions of those they interact with. Remembering and reasoning over others' mental states ensures efficient communication by avoiding having to repeat information, and in cases where cooperation is involved contributes to achieving a common goal with minimal effort.

Because of this, there is growing interest in developing agents that can exhibit this kind of behavior, referred to as Theory of Mind (ToM) by developmental psychologists (Premack & Woodruff, 1978).[1] Previous work on agents imbued with some capability of ToM has focused mainly on two types of tasks. The former are tasks where the agent is a passive observer of a scene that has to predict the future by reasoning over others' mental states. These tasks may involve natural language (Nematzadeh et al., 2018) or be purely spatial (Gandhi et al., 2021; Rabinowitz et al., 2018; Baker et al., 2011). The latter are tasks where the ToM agent has a specific role, such as "the speaker" in speaker-listener scenarios (Zhu et al., 2021).

In contrast, human cooperation and communication is very often multi-party, and rarely assumes that people have pre-fixed roles. Moreover, human interlocutors are seldom passive observers of a scene but instead proactively interact with their environment. Since previous domains limited us to research questions where most parties involved did not have an active role, we developed a more flexible environment where we can now study what happens when all participants must act as both speaker and listener. In this paper, we present SymmToM, a fully symmetric multi-agent environment where all agents can see, hear, speak, and move, and are active players of a simple information-gathering game. To solve SymmToM, agents need to exhibit different levels of ToM, as well as efficiently communicate through a simple channel with a fixed set of symbols.

---

[1]In the present work we focus solely on reasoning over mental states. Other aspects of ToM include understanding preferences, goals, and desires of others. Multi-agent scenarios for inferring agent's goals have been studied (Ullman et al., 2009), and passive-observer benchmarks (Gandhi et al., 2021; Shu et al., 2021; Netanyahu* et al., 2021) have been proposed for evaluating understanding of agent's goals and preferences.

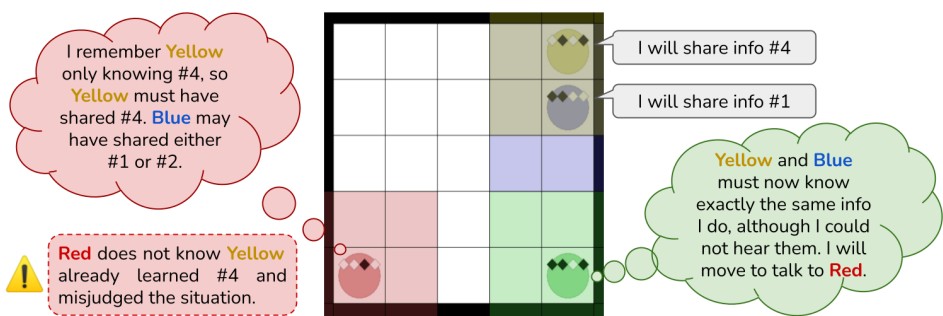

Figure 1: In SymmToM, agents aim to gain all the information available (depicted as diamonds, black for known information, white for unknown). Since hearing is limited to its neighbor cells, they must guess what happened beyond this range. Agents can see the whole grid, but even then, mistakes in inferences may happen (as it did in this example to the red agent).

SymmToM is a partially observable setting for all agents: even when agents have full vision, hearing may be limited. This also differentiates SymmToM from prior work, as modeling may require *probabilistic theory of mind*. In other words, agents need to not only remember and infer other agents' knowledge based on what they saw, but also estimate the probability that certain events happened. This estimation may be performed by assuming other agents' optimal behavior and processing the partial information available. Despite its simplicity, SymmToM fulfills the properties required for symmetric ToM to arise, which will be discussed in the following section.

We find that SymmToM cannot be completely solved neither by using well-known multi-agent deep reinforcement learning (RL) models, nor by tailoring those models to our task. We show that even maintaining the simple rules of the environment, modifying its parameters results in much more difficult challenges, even for models where we artificially introduce perfect information. We discuss examples where different levels of theory of mind are required to solve the task, and possible metrics.

## 2 THEORY-OF-MIND AGENTS

A belief Theory-of-Mind agent can be defined as a modification of the standard multi-agent RL paradigm, where the agents' policies are conditioned on their beliefs about others. Formally, we define a reinforcement learning problem $\mathcal{M}$ as a tuple of a state space $\mathcal{S}$, action space $\mathcal{A}$, state transition probability function $T \in \mathcal{S} \times \mathcal{A} \to \mathbb{R}$, and reward function $R \in \mathcal{S} \times \mathcal{A} \to \mathbb{R}$, i.e. $\mathcal{M} := \langle \mathcal{S}, \mathcal{A}, T, R \rangle$. In this setting, an agent learns a (possibly probabilistic) policy $\pi : \mathcal{S} \to \mathcal{A}$ that maps from states to actions, with the goal of maximizing reward.

In a multi-agent RL setting each agent can potentially have its own state space, action space, transition probabilities, and reward function, so we can define an instance of $\mathcal{M}_i = \langle \mathcal{S}_i, \mathcal{A}_i, T_i, R_i \rangle$ for each agent $i$. For convenience, we can also define a joint state space $\mathcal{S} = \bigcup_i \mathcal{S}_i$ that describes the entire world in which all agents are interacting. Importantly, in this setting each agent will have its own view of the entirety of the world, described by a conditional observation function $\omega_i : \mathcal{S} \to \Omega_i$ that maps from the state of the entire environment to only the information observable by agent $i$.

As elaborated above, ToM is the ability to know (and act upon) the knowledge that an agent has. Agents with *no ToM* will follow a policy that depends only on their current (potentially partial or noisy) observation of their environment: $\pi_i(a_{i,t} \mid \omega_i(s_t))$. Agents with *zeroth order ToM* can reason over their own knowledge. These agents will be stateful, $\pi_i(\cdot \mid \omega_i(s_t), h_t^{(i)})$, where $h_t^{(i)}$ is $i$'s hidden state. Hidden states are always accessible to their owner, i.e. $i$ has access to $h_t^{(i)}$.

Agents with capabilities of reasoning over other agents' mental states will need to estimate $h_t^{(j)}$ for $j \neq i$. We will denote the estimation that $i$ does of $j$'s mental state in time $t$ as $\hat{h}_t^{(i,j)}$:

$$\pi_i(\cdot \mid \omega_i(s_t), h_t^{(i)}, \hat{h}_t^{(i,1)}, \ldots \hat{h}_t^{(i,i-1)}, \hat{h}_t^{(i,i+1)} \ldots \hat{h}_t^{(n)})$$

How do we estimate $\hat{h}_t^{(i,j)}$? As a function of $i$'s (the predicting agent) previous hidden state $t-1$, $i$'s observation in $t-1$, and $i$'s prediction of the hidden states of every agent in the previous turn:

$$\hat{h}_t^{(i+1)} = f(h_{t-1}^{(i)}, \omega_i(s_{t-1}), \hat{h}_{t-1}^{(i,1)}, \ldots \hat{h}_{t-1}^{(i,i-1)}, \hat{h}_{t-1}^{(i,i+1)} \ldots \hat{h}_{t-1}^{(i,n)})$$

$i$'s prediction of other agents' observation in $t-1$ is also crucial, but not explicitly mentioned since it can be computed using $\omega_i(s_{t-1})$. For the initial turn, $\hat{h}_0^{(i,j)}$ may be initialized differently depending on the problem: if initial knowledge is public, $\hat{h}_0^{(i,j)}$ is trivial; if not, $\hat{h}_0^{(i,j)}$ may be estimated.

## 3 SYMMETRIC THEORY-OF-MIND

We define symmetric theory of mind environments as settings where theory of mind is required to perform a task successfully, and all agents have the same abilities. There are at least four defining characteristics of symmetric ToM to arise:

**Symmetric action space.** In symmetric ToM all agents are required to have the same action space (in contrast to, for example, ToM tasks in speaker-listener settings). Concretely, $\mathcal{A}_i = \mathcal{A}_j \neq \emptyset \; \forall i, j$.

**Imperfect information.** In perfect information scenarios all knowledge is public, making it impossible to have agents with different mental states. In ToM tasks in general, there could be a subset of agents with perfect information: one example would a passive observer that needs to predict future behavior. In symmetric ToM, since all agents have the same abilities and roles, all agents must have imperfect information. More precisely, $\omega_i$ must not be the identity for any agent $i$.

**Observation of others.** Agents must have at least partial information of another agent to estimate its mental state. In contrast to passive-observer settings, in symmetric ToM every agent must be able to partially observe all others. More precisely, $\omega_i$ must observe at least partial information about $s_t^{(j)}$ (the subset of $s_t$ that refers to agent $j$), although we do not require $s_t^{(j)} \neq \emptyset$ in every single turn. Moreover, if communication is allowed, it is desirable to partially observe or infer interactions between two or more agents to develop second order ToM (i.e. predicting what an agent thinks about what another agent is thinking) or higher.

**Information-seeking behavior.** It should be relevant for successfully performing the task to gather as much information as possible, and this information-gathering should involve some level of reasoning over other agent's knowledge. This is true for first-order ToM tasks in general, and could be formalized as $\pi^* \neq \pi$ for any zeroth-order ToM policy $\pi_i(\cdot \mid \omega_i(s_t), h_t^{(i)})$. Furthermore, it would be desirable to design a task with **perpetual** information seeking behavior, since it would ensure that all agents have an incentive to play efficiently even in long episodes. If one wants to design a task with perpetual information-seeking and finite knowledge, information must be forgotten eventually. A forgetting mechanism could be implemented as an explicit loss of knowledge under specific conditions, or make remembrances less reliable or noisy. Moreover, this introduces the concept of *information staleness*. Since information is not cumulative and the environment is only partially observable, agents will need to estimate whether what they knew to be true still holds in the present.

## 4 THE SYMMTOM ENVIRONMENT

SymmToM is an environment where $a$ agents are placed in a $w \times w$ grid world, and attempt to maximize their reward by gathering all the information available in the environment. There are $c$ available information pieces, that each agent may or may not know initially. Information pieces known at the start of an episode are referred to as *first-hand information*. Each turn, agents may move through the grid to one of its four neighboring cells, and may speak exactly one of their currently known information pieces. More precisely, the action space of agent $j$ is defined as follows:

$$\mathcal{A}_j = \{left, right, up, down, no \; movement\} \times \{1, \ldots, c\} \tag{1}$$

When an agent utters an information piece, it is heard by every agent in its hearing range (an $h \times h$ grid centered in each agent, with $h < 2w - 1$). The agents who heard the utterance will be able

**a.** 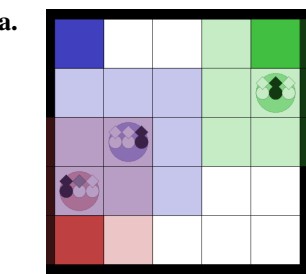 **b.** 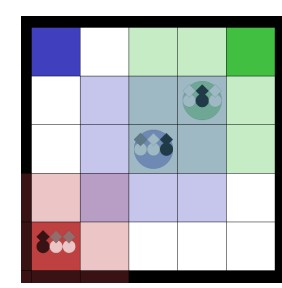 **c.** 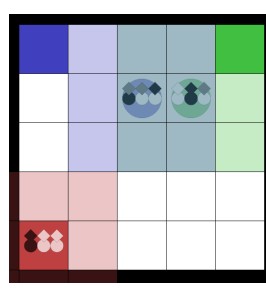

Figure 2: *Example of three consecutive turns in an episode.* There are three agents in a $5 \times 5$ grid, each with a hearing range of $1$ (shaded in the same color of the agent). Fully-colored cells depict recharge bases. Information is represented by diamonds: black, gray, and white diamonds represent an information piece known first-hand, second-hand, and not known, respectively. Black circles show the information piece currently being said by each agent.

to share this newly-learned information with others in following turns. We refer to this as *second-hand information*, since it is learned –as opposed to *first-hand information*, given at the start of each episode. The state space is comprised of the position of the agents and their current knowledge:

$$\mathcal{S} = \{\{(p_i, k_i), \text{ for } i \in \{1, \ldots, a\}\} \text{ where } p_i \in \{1, \ldots, w\} \times \{1, \ldots, w\}, \text{ and } k_i \in \{0, 1\}^c\}$$

Each agent aims to maximize their individual reward $\mathcal{R}_i$ via information seeking and sharing. Rewards are earned by hearing a new piece of information, giving someone else a new piece of information, or correctly using *recharge bases*. Recharge bases are special cells where agents can reset their knowledge in exchange for a large reward (e.g. $(n-1)c$ times the reward for listening to or sharing new information). Each agent has its own stationary recharge base during an episode. To trigger a base, an agent must step into its designated base having acquired all the available pieces of information, causing the agent to lose all the second-hand information it learned. Recharge bases guarantee that there is always reward to seek in this environment. Concretely, if $s = \{(p_i, k_i), \text{ for } i \in \{1, \ldots, a\}\}$ and $a_i = (a_i^{\text{dir}}, a_i^{\text{comm}})$, we can define the reward as the addition of the reward for hearing new information, giving new information, and using the recharge base:

$$R_i(s, a_i) = \sum_{i \neq j} \mathbb{1}\{||p_i - p_j||_\infty \leq h \text{ and } k_{i, a_j^{\text{comm}}} = 0\} + \sum_{i \neq j} \mathbb{1}\{||p_i - p_j||_\infty \leq h \text{ and } k_{j, a_i^{\text{comm}}} = 0\}$$
$$+ (n-1) \cdot c \cdot \mathbb{1}\{p_i = \text{base}_i \text{ and } k_j = \{1, 1, \ldots, 1\}\}$$

A non-ToM agent can have only limited success in this environment. Without reasoning about its own knowledge (i.e. without *zeroth order ToM*), it does not know when to use a recharge base. Moreover, without knowledge about other agent's knowledge (i.e. without *first order ToM*) it is not possibly to know which agents possess the information pieces it is lacking. Even if it accidentally hears information, a non-first-order ToM agent cannot efficiently decide what to utter in response to maximize its reward. Higher order ToM is also often needed in SymmToM, as we will discuss further in Section 7.3.

Even though we only discussed a collaborative task for SymmToM, it can easily be extended for competitive tasks. Moreover, all our models are also designed to work under competitive settings. SymmToM satisfies the desiderata we laid out in the previous section, as we will detail below:

**Symmetric action space.** As defined in Eq. 1, $\mathcal{A}_i = \mathcal{A}_j$ for all $i, j$. Only a subset may be available at a time since agents cannot step outside the grid, speak a piece they have not heard, or move if they would collide with another agent in the same cell, but they all share the same action space.

**Imperfect information.** Messages sent by agents outside of the hearing range will not be heard. For example, in Fig. 2a green sends a message but it is not heard by anyone, since it is outside of red's and blue's range. Hearing ranges are guaranteed not to cover the whole grid, since $h < 2w - 1$.

**Observation of others.** Agents have perfect vision of the grid, even if they cannot hear what was said outside of their hearing range. Hence, an agent may see that two agents were in range of each

other, and thus probably interacted, but not hear what was communicated. An example of this can be seen in Fig. 2a, where green observes blue and red interacting without hearing what was uttered.

The uncertainty in the observation also differentiates SymmToM from prior work: to solve the task perfectly, an agent needs to assess the probability that other agents outside its hearing range shared a specific piece of information to avoid repetition. This estimation may be performed using the knowledge of what each agent knows (first order ToM), the perceived knowledge of each of the agents in the interaction (second order ToM), as well as higher order ToM.

**Information-seeking behavior**   Rewards are explicitly given for hearing and sharing novel information, guaranteeing information-seeking is crucial in SymmToM. Recharge bases ensure that the optimal solution is not for all agents to accumulate in the same spot and quickly share all the information available; and that the information tracking required is more complex than accumulating past events. Conceptually, with recharge bases we introduce an explicit and observable forgetting mechanism. As discussed in Section 3, this allows for perpetual information seeking and requires information staleness estimation. An example of successful recharge base use is shown in Fig. 2b.

## 5    BASELINE LEARNING ALGORITHMS AND OTHER BOUNDS

To learn a policy for acting in the multi-agent SymmToM environment, it is a good strategy to use a multi-agent reinforcement learning algorithm. We use MADDPG (Lowe et al., 2017), a well-known multi-agent actor-critic framework with centralized planning and decentralized execution, to counter the non-stationarity nature of multi-agent settings. In MADDPG, each actor policy receives its observation space as input, and outputs the probability of taking each action.

Notably, actors in MADDPG have no mechanism for remembering past turns. This is a critical issue in SymmToM, as agents cannot remember which pieces they currently know, which ones they shared and to whom, and other witnessed interactions. To mitigate this, it is necessary to add a recurrence mechanism to carry over information from past turns. One option would be to modify the agent policy using a recurrent network like an LSTM, as RMADDPG (Wang et al., 2020) does.

**Perfect Information, Heuristic and Lower Bound Models**   Performance is difficult to interpret without simpler baselines. As a lower bound model we use the original MADDPG, that since it does not have recurrence embedded, should perform worse or equal to any of the modifications described above. We also include an oracle model (*MADDPG-Oracle*), that does not require theory of mind since it receives the current knowledge $K$ for all agents in its observation space. The performance of MADDPG-Oracle may not always be achieved, as there could be unobserved communication with multiple situations happening with equal probability. Moreover, as the number of agents and size of the grid increases, current reinforcement learning models may not be able to find an optimal spatial exploration policy; they may also not be capable of inferring the optimal piece of information to communicate in larger settings. In these cases, MADDPG-Oracle may not perform optimally, so we also include a baseline with heuristic agents to compare performance.

Heuristic agents will always move to the center of the board and communicate round-robin all the information pieces they know until they have all the available knowledge. Then, they will move efficiently to their recharge base and come back to the center of the grid, where the process restarts. We must mention that this heuristic is not necessarily the perfect policy, but it will serve as a baseline to note settings where current MARL models fail even with perfect information. Qualitatively, smaller settings have shown to approximately follow a policy like the heuristic just described.

## 6    DIRECT MODELING OF SYMMETRIC THEORY OF MIND

In contrast to RMADDPG (Wang et al., 2020), we specifically design algorithms for our environment to maximize performance. Intuitively, our model computes a matrix, $K \in \{0, 1\}^{c \times a}$, that reflects the information pieces known by each agent from the perspective of the agent being modeled: $K_{ij}$ reflects if the agent being modeled believes that agent $j$ knows $i$. $K$ is updated every turn and used as input of the following turn of the agent, obtaining the desired recurrent behavior. $K$ is also concatenated to the usual observation space, to be processed by a two-layer ReLU MLP and

obtain the probability distributions for speech and movement, as in the original MADDPG. There are several ways to approximate $K$. It is important to note that each agent can only partially observe communication, and therefore it is impossible to perfectly compute $K$ deterministically.

The current knowledge is comprised of first-hand information (the initial knowledge of every agent, $F$, publicly available) and second-hand information. Second-hand information may have been heard this turn ($S$, whose computation will be discussed below) or in previous turns (captured in the $K$ received from the previous turn, noted $K^{(t-1)}$). Additionally, knowledge may be forgotten when an agent steps on a base having all the information pieces. To express this, we precompute a vector $B \in \{0,1\}^a$ that reflects whether each agent is currently on its base; and a vector $E \in \{0,1\}^a$ that determines if an agent is entitled to use their recharge base: $E_j = \mathbb{1}_{\sum_i K_{ij}=c}$ for all $j \in \{1, \ldots, a\}$. We are then able to compute $K$ as follows:

$$K_{ij}^{(t)} = (F_{ij} \text{ or } S_{ij} = 1 \text{ or } K_{ij}^{(t-1)} = 1) \text{ and not } (B_{ij} \text{ and } E_{ij}) \tag{2}$$

$F$, $K^{(t-1)}$, and $B$ are given as input, but we have not yet discussed the computation of the second-hand information $S$. $S$ often cannot be deterministically computed, since our setting is partially observable. We will identify three behaviors and then compute $S$ as the sum of the three:

$$S = S^{[0]} + S^{[1]} + S^{[2]}$$

For simplicity, we will assume from now on that we are modeling agent $k$. $S^{[0]}$ will symbolize the implications of the information spoken by agent $k$: if agent $k$ speaks a piece of information, they thus know that every agent in its hearing range must have heard it (first order ToM). $S^{[1]}$ will symbolize the implications of information heard by $k$: this includes updating $k$'s known information (zeroth order ToM) and the information of every agent that is also in hearing range of the speaker heard by $k$. $S^{[2]}$ will symbolize the estimation of information pieces communicated between agents that are out of $k$'s hearing range. Since we assume perfect vision, $k$ will be able to see if two agents are in range of each other, but not hear what they communicate (if they do at all).

$S^{[0]}$ and $S^{[1]}$ can be deterministically computed. To do so, it is key to note that every actor knows the set of actions $A \in \{0,1\}^{c \times a}$ performed by each agent last turn, given that those actions were performed in their hearing range. Moreover, each agent knows which agents are in its range, as they all have perfect vision. We precompute $H \in \{0,1\}^{a \times a}$ to denote if two given agents are in range.

Then, $S_{ij}^{[0]} = 1$ if and only if information piece $i$ was said by $k$, and agents $k$ and $j$ are in hearing range of each other. More formally,

$$S_{ij}^{[0]} = A_{ik} \cdot H_{kj}$$

$S_{ij}^{[1]} = 1$ if and only if agent $k$ (the actor we are modeling) heard some agent $\ell$ speaking information piece $i$, and agent $j$ is also in range of agent $\ell$. Note that agent $k$ does not need to be in hearing range of agent $j$. More precisely,

$$S_{ij}^{[1]} = A_{i\ell} \cdot H_{k\ell} \cdot H_{\ell j}, \text{ for any agent } \ell$$

$S^{[2]}$ –the interactions between agents not in hearing range of the agent we are modeling– can be estimated in different ways. A conservative approach would be to not estimate interactions we do not witness ($S^{[2]} = 0$, which we will call *MADDPG-ConservativeEncounter (MADDPG-CE)*); and another approach would be to assume that every interaction we do not witness results in sharing a piece of information that will maximize the rewards in that immediate turn. We will call this last approach *MADDPG-GreedyEncounter (MADDPG-GE)*. MADDPG-GE assumes agents play optimally, but does not necessarily know all the known information and that could lead to a wrong prediction. This is particularly true during training, as agents may not behave optimally. The computation of $S^{[2]}$ for MADDPG-GE is as follows.

First, we predict the information piece $U_\ell$ that agent $\ell$ uttered. MADDPG-GE predicts $U_\ell$ will be the piece that the least number of agents in range know, as it will maximize immediate reward:

$$U_\ell = \arg\min_i \sum_j (K_{ij} \text{ and } H_{j\ell}) \in \{1, \ldots c\}$$

With this prediction, agent $j$ will know information $i$ if at least one agent in its range said it:

$$S_{ij}^{[2]} = 1 \text{ if exists } \ell \neq k \text{ such that } U_\ell = j \text{ and } H_{j\ell} \text{ and } j \neq k \text{ else } 0$$

**MADDPG-EstimatedEncounter (MADDPG-EE)**   MADDPG-CE and MADDPG-GE are two paths to information sharing estimation, but in none of them do we estimate the probability of an agent knowing a specific piece of information. In *MADDPG-EstimatedEncounter (MADDPG-EE)*, known information of other agents is not binary, i.e. $K_{ij} \in [0, 1]$. This added flexibility can avoid making predictions of shared information based upon unreliable information.

MADDPG-EE estimates the probability that an agent $j$ uttered each piece of information ($U_j \in \mathbb{R}^c$) by providing the current information of all agents in its range to an MLP:

$$U_j = \mathrm{softmax}(f(K_{1j}, \ldots, K_{cj}, \{K_{1\ell}, \ldots, K_{c\ell} \text{ for all } \ell \text{ where } H_{j\ell}\})), \text{ with } f \text{ an MLP}$$

Then, the probability of having heard a specific piece of information will be the complement of not having heard it, which in turn means that none of the agents in range said it. More formally,

$$S_{ij}^{[2]} = 1 - \prod_{\ell, H_{j\ell}=1} 1 - U_\ell$$

Since MADDPG-EE requires functions to be differentiable, we use a differential approximation of Eq. 2. A pseudocode of MADDPG-EE's implementation can be found in Section A.4. MADDPG-EE solely focuses on first order ToM, and we leave to future work modeling with second order ToM. The structure of the model would be similar but with an order of magnitude more parameters.

## 7    EXPERIMENTS

### 7.1    EXPERIMENTAL SETTINGS

In this section, we compare the different algorithms explained in the previous section. The observation space will be constituted of a processed version of the last turn in the episode, to keep the input size controlled. More precisely, the observation space is composed of: the position of all agents, all recharge bases, the current direction each agent is moving towards to and what they communicated in the last turn, the presence of a wall in each of the immediate surroundings, and every agents' first-hand information. First-hand information is publicly available in our experiments to moderate the difficulty of the setup, but this constraint could also be removed. This simple setting is still partially observable, since the agents cannot hear interactions outside of their hearing range.

We use the reward as our main evaluation metric. This metric indirectly evaluates ToM capabilities, since information-seeking is at the core of SymmToM. We train through 60000 episodes, and with 7 random seeds to account for high variances in the rewards obtained. Our policies are parametrized by a two-layer ReLU MLP with 64 units per layer, as in the original MADDPG (Lowe et al., 2017). MADDPG-EE's function $f$ is also a two-layer ReLU MLP with 64 units per layer.

We test two board sizes ($w = 6$, and $w = 12$), two numbers of agents ($a = 3$ and $a = 4$), and three quantities of information pieces ($c = a$, $c = 2a$ $c = 3a$). The length of each episode is set to $5w$. More detail about design decisions can be found in Section A.5.

### 7.2    MAIN RESULTS

As we can observe in Table 1, there is a significant difference in performance between MADDPG-Oracle and MADDPG (MADDPG-Oracle is 123% better on average): this confirms that developing ToM and recurrence is vital to perform successfully in SymmToM. MADDPG-Oracle is often not an upper bound: when $c > a$, the heuristic performs better (101% on average). This shows that even with perfect information, it can be difficult to learn the optimal policy using MADDPG. Moreover, models with a recurrence mechanism perform significantly better than MADDPG (61% better on average), also showing that remembering past information gives a notable advantage. As expected, having recurrent models tailored to our problem resulted in better performance than a general LSTM recurrence (RMADDPG). The performance of the best of the tailored models (MADDPG-CE, MADDPG-GE, MADDPG-EE) was 42% better on average than plain RMADDPG. LSTM was able to surpass the best of the tailored models only for $a = 3, w = 12, c = 3a$.

Increasing $c$ generally decreases global rewards for learned agents (on average, rewards for $c = 2a$ are 75.54% of those for $c = a$, and rewards for $c = 3a$ are 75.66% of those for $c = a$).

Table 1: Average rewards per agent in trained models evaluated during 1000 episodes. 7 runs are averaged for each, using the best checkpoint: this compensates for collapses in performance seen in Fig. 5 and Fig. 6. Values shown are individual rewards to normalize by the number of agents. Bold lettering represents the best result of a learned imperfect-information model for each setting. Standard deviations are detailed in A.5.

| agents ($a$) | 3 | | | | | | 4 | | | | | |
|---|---|---|---|---|---|---|---|---|---|---|---|---|
| grid width ($w$) | 6 | | | 12 | | | 6 | | | 12 | | |
| info pieces ($c$) | $a$ | $2a$ | $3a$ | $a$ | $2a$ | $3a$ | $a$ | $2a$ | $3a$ | $a$ | $2a$ | $3a$ |
| Heuristic | 38 | 54 | 58 | 37 | 57 | 72 | 60 | 73 | 74 | 58 | 89 | 101 |
| MADDPG-Oracle | 42 | 49 | 39 | 54 | 38 | 41 | 71 | 45 | 30 | 57 | 33 | 29 |
| MADDPG | 39 | 18 | 16 | 35 | 32 | 11 | 34 | 23 | 13 | 18 | 14 | 14 |
| +RNN (RMADDPG) | 37 | 19 | 19 | 44 | 27 | **19** | 29 | 20 | 16 | 32 | 17 | 16 |
| +Conservative (CE) | 36 | **26** | 33 | 46 | 43 | 14 | **40** | 30 | **25** | 27 | 27 | 16 |
| +Greedy (GE) | 34 | 25 | **33** | **47** | **68** | 15 | 35 | **30** | 25 | **35** | **29** | **17** |
| +Estimated (EE) | **40** | 19 | 15 | 39 | 26 | 11 | 36 | 22 | 14 | 25 | 19 | 15 |

This suggests that probabilistic decisions are harder to learn, or impossible to successfully navigate when several events are equally likely. MADDPG-EE did not show improvements over the other learned agents, and in some cases decreased its performance more heavily than other learned agents (e.g. $w = 6$, $c = 3a$). MADDPG-EE uses an MLP in its definition of $S^{[2]}$, which gives more flexibility but also implies a more complex function to learn. We leave to future work to explore other probabilistic agents, but the significant difference in performance between all of the learned models and the highest performing ones (MADDPG-Oracle and the heuristic) shows there is ample space for improvement in this task, and hence proves SymmToM to be a simple yet unsolved benchmark.

Increasing $a$ results in a 10% reduction of performance on average for learned models. Nonetheless, the heuristic improved its rewards by an average of 46%, given the larger opportunities for rewards when including an additional listener. Overall, this implies that increasing $a$ also makes the setup significantly more difficult. Finally, increasing $w$ did not have a conclusive result: for $a = 4$ it consistently decreased performance in ∼17%, but for $a = 3$ we saw an improvement of 16% and 61% for $c = a$ and $c = 2a$ respectively.

In sum, modifying $c$ and $a$ provides an easy way of making a setting more difficult without introducing additional rules.

## 7.3 DISCUSSION

In the past section we analyzed results using the metric of episode rewards. Although the rewards in SymmToM are designed to correlate with information seeking and knowledge state of the agent themselves and others, they do not show explicitly if agents are exhibiting theory of mind. To do so more directly, we develop two categories of possible analyses: scenarios specifically designed to test theory of mind, and post-hoc analyses of episodes.

A classic example of a scenario specifically designed to test ToM behavior is the Sally-Anne task (Wimmer & Perner, 1983). This false belief task, originally designed for children, aims to test if a passive observer can answer questions about the beliefs of another person, in situations where that belief may not match reality. If we were to use it for machine ToM, we could repeat the experiment and ask an agent to predict the position of an object varying the underlying conditions. This test is feasible because there is only one agent with freedom of action, which ensures that desired conditions are met every time. Testing becomes unfeasible when giving multiple agents freedom of action, as constraints planned in test design may be broken by a collective drift from the strategy thought by the designer. Testing becomes easier if we allow for controlling all agents but one, as shown in Fig. 3. Other tests besides the ones shown may be designed. In particular, in Fig. 3d we show an example of probabilistic ToM where two communicative events are equally likely, but one could modify this scenario to have different probabilities and test the expected value of the turns until red successfully shares an information piece. One could also design *retroactive deduction* tests: for example, in Fig. 3d if red communicates and receives no reward, it can deduce that green had

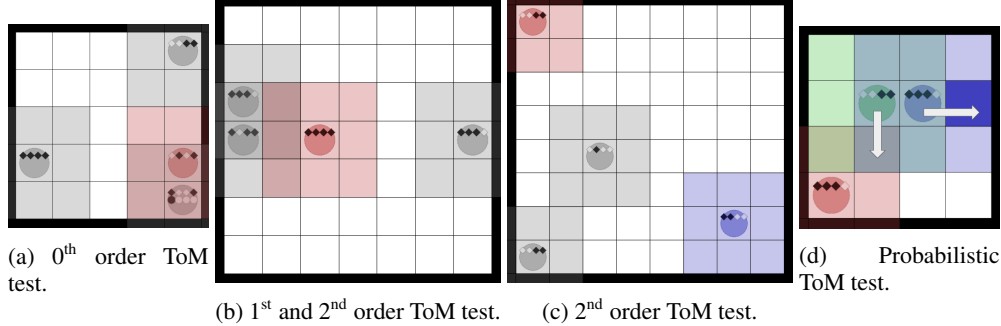

(a) $0^{th}$ order ToM test.

(b) $1^{st}$ and $2^{nd}$ order ToM test.

(c) $2^{nd}$ order ToM test.

(d) Probabilistic ToM test.

Figure 3: *Example of tests for zeroth, first, second order, and probabilistic ToM.* We test red agents, immobilize gray agents, and control blue and green agents' movements. In Fig. 3a, red will go to the top right if it remembers to have heard the first piece, and to the left otherwise. In Fig. 3b, red will move to the right if and only if it assumes that the two agents on the left played optimally (red cannot hear what they communicated). In Fig. 3c, blue is controlled to ensure it will search the agent on the bottom left (its optimal play, in five moves). Red's optimal move is to meet blue, and hence must only move to the bottom left, even if the agent currently there will not provide any reward. In Fig. 3d, red will interact with green not knowing what it communicated with blue in the previous turn. It should be able to communicate the missing piece with an expected value of 1.5 turns.

received that information from blue. If there had been another agent (let's say, a yellow agent) in range of blue when it spoke to green, the red agent could also update its knowledge about yellow. Results for the tests proposed in Figure 3 are detailed in Appendix A.1.

Post-hoc analysis also has its challenges in multi-agent settings, even in the most direct cases. Thanks to our reward shaping, using recharge bases is always the optimal move when an agent has all the information available: an agent will have a reward of $(n-1)c$ for using the base, whereas it can only gain up to $n-1+c-1$ per turn if it decides not to use it. Even in this case, small delays in using the base may occur, for example if the agent can gather additional rewards on its path to the base. More generally, having multiple agents makes a specific behaviors attributable to any of the several events happening at once, or a combination of them.

Even though it may be difficult to establish causality when observing single episodes, we developed metrics that comparatively show which models are using specific features of the environment better than others. Examples of metrics are unsuccessful recharge base usage count; number of times an agent shares an information piece everyone in its range already knows when having better alternatives; number of times an agent moves away from every agent when not having all the information pieces available; among others. See Appendix A.2 for detailed description and results. Reward can also be understood as a post-hoc metric with a more indirect intepretation.

Post-hoc analyses of single episodes can also be blurred by emergent communication. Because agents were trained together, they may develop special meaning assignment to particular physical movements or messages. Even though qualitatively this does not seem to be the case for the models presented in the paper, tests should also account for future developments. This also implies that one should not overinterpret small differences in the metrics described in the paragraph above.

## 8 CONCLUSIONS AND FUTURE WORK

We defined a framework to analyze machine theory of mind in a multi-agent symmetric setting, a more realistic setup than the tasks currently used in the community. Based on the four properties needed for symmetric theory of mind to arise, we provided a simplified setup on which to test the problem, and we showed we can easily increase difficulty by growing the number of agents or communication pieces. Our main goal in this work was not to solve symmetric theory of mind, but rather to give a starting point to explore more complex models in this area. We showed that even with this minimal set of rules, SymmToM proves algorithmically difficult for current multi-agent deep reinforcement learning models, even when tailoring them to our specific task. We leave to

future work to develop models that handle second-order theory of mind and beyond, and models that periodically reevaluate past turns to make new deductions with information gained *a posteriori* (i.e., models that pass retroactive deduction tests). Another interesting direction would be to replace the information pieces with constrained natural language: communication sharing in our task is binary, whereas in language there is flexibility to communicate different subsets of a knowledge base using a single sentence. We will make our codebase public upon publication, that also includes additional observation space restrictions to increase difficulty.

## ETHICS AND REPRODUCIBILITY STATEMENT

Theory of mind research at its core deals with understanding the mental states of other individuals. In the present work we focused on collaborative machine theory of mind, which entails interactions only between artificial agents, and only in scenarios where every party involved has the same incentive structure. This design decision is intentional. Other approaches to theory of mind research could include scenarios with human-agent interaction, which could potentially lead to agents learning to model human players' mental states. This is not concerning per se, but in scenarios where players do not have the same incentive structure, it could lead to agents learning to deceive other players (potentially human players). The state of the art in machine theory of mind is still far away from these capabilities, but we believe that experiment design choices should always take this matter into account.

Regarding reproducibility, we will make all code public upon acceptance, including the environment and the models' code. The exact set of parameters used for training will also be shared. Multi-agent reinforcement learning models do have high variance, so models should be run several times to see similar confidence intervals as the ones shown in Fig. 5 and Fig. 6.

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

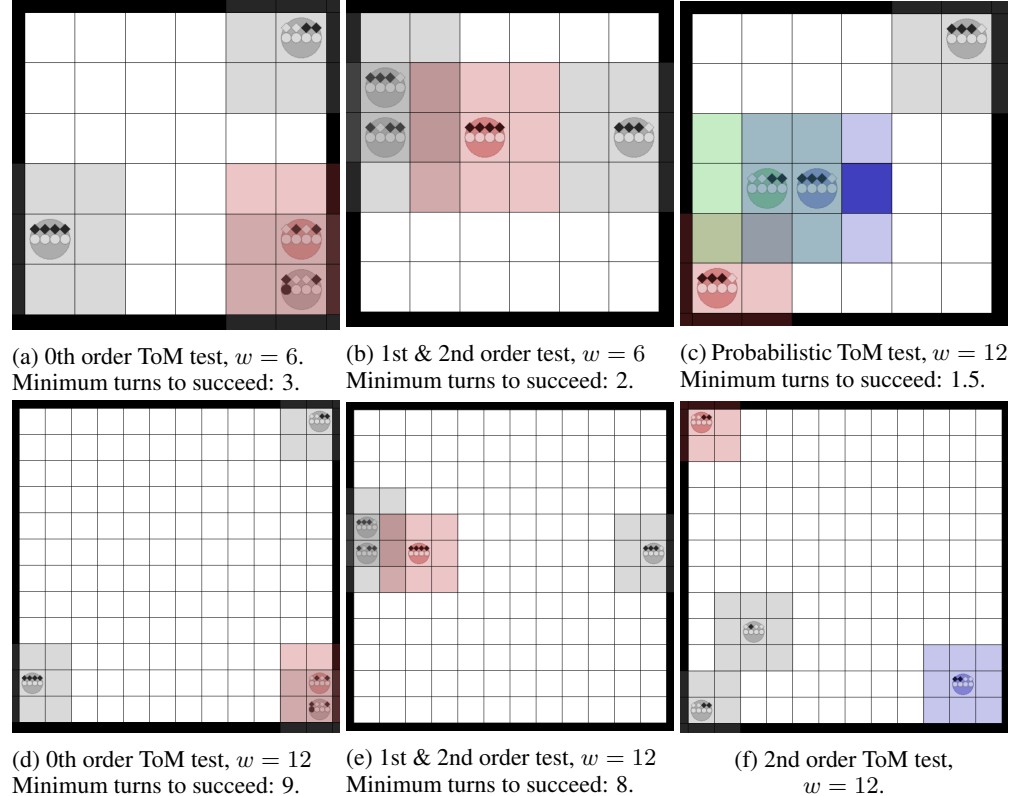

(a) 0th order ToM test, $w = 6$.
Minimum turns to succeed: 3.

(b) 1st & 2nd order test, $w = 6$
Minimum turns to succeed: 2.

(c) Probabilistic ToM test, $w = 12$
Minimum turns to succeed: 1.5.

(d) 0th order ToM test, $w = 12$
Minimum turns to succeed: 9.

(e) 1st & 2nd order test, $w = 12$
Minimum turns to succeed: 8.

(f) 2nd order ToM test,
$w = 12$.

Figure 4: Depictions of rescaled tests from Figure 3, designed to match some of the parameter combinations already experimented on.

# A   APPENDIX

## A.1   AD-HOC THEORY OF MIND TESTS

We test on the four examples shown in Figure 3, adapting the examples to fit one of the grid sizes we already experimented on. For the tests described in Figure 3a and Figure 3b, we test two different grid sizes: $w = 6$ and $w = 12$. For the tests described in Figure 3c and Figure 3d we only test $w = 12$ and $w = 6$ respectively. Image depictions of the exact test configurations can be seen in Figure 4.

We measure three metrics: *average success rate*, *average failure rate*, and *ratio of average turns to succeed vs. optimum (RATSO)*. Note that Average Success Rate and Average Failure Rate do not necessarily sum 1 since these two metrics only include trials where the agent reached any of the two proposed outcomes. If, for example, the agent never moved from the starting point, the trial would not be counted positively towards Avg. Success Rate or Avg. Failure Rate. In addition, *ratio of average turns to succeed vs. optimum (RATSO)* is the ratio between the average turns it took to succeed in successful trials, and the optimum number of turns to succeed in a specific trial.

For the tests in Figure 4a, 4b, 4d, and 4e, the trial ends when the red agent reaches the hearing range of one of the two possible target agents. The test depicted in Figure 4f is a pass/fail test: if red moves suboptimally at any point before meeting blue, the trial is declared as failed. This makes it a particularly difficult test to pass at random. Because of the nature of this second order ToM test, we only report the average success rate. Finally, for the probabilistic ToM test we want to measure how fast can red communicate all the information it has to green. The optimal number of turns is 1.5 (as discribed in Figure 3), and because of the nature of this test we will only report RATSO.

All results can be found in Table 2. As expected, a larger average success rate correlates with higher reward models (MADDPG-CE and MADDPG-GE are the best models), suggesting that the

| ToM test | Fig. 3a (0th order) | | | | | | | | Fig. 3c (2nd order) |
|---|---|---|---|---|---|---|---|---|---|
| grid width ($w$) | 6 | | | | 12 | | | | 12 |
| | SR | FR | 1-SR-FR | RATSO | SR | FR | 1-SR-FR | RATSO | SR |
| MADDPG | 17% | 75% | 9% | 3.66 | 1% | 44% | 55% | 3.75 | 33% |
| RMADDPG | 18% | 59% | 24% | 5.30 | 1% | 3% | 96% | 5.39 | 32% |
| MADDPG-CE | 32% | 49% | 19% | 4.94 | 10% | 1% | 89% | 5.00 | 24% |
| MADDPG-GE | 32% | 40% | 28% | 4.97 | 10% | 12% | 78% | 4.99 | 32% |
| MADDPG-EE | 16% | 54% | 29% | 5.26 | 0% | 25% | 75% | 4.39 | 33% |

| ToM test | Fig. 3b (1st and 2nd order) | | | | | | | | Fig. 3d (probabilistic) |
|---|---|---|---|---|---|---|---|---|---|
| grid width ($w$) | 6 | | | | 12 | | | | 6 |
| | SR | FR | 1-SR-FR | RATSO | SR | FR | 1-SR-FR | RATSO | RATSO |
| MADDPG | 9% | 79% | 12% | 3.37 | 0% | 62% | 38% | 5.49 | 6.40 |
| RMADDPG | 18% | 74% | 8% | 3.70 | 2% | 61% | 37% | 4.49 | 5.02 |
| MADDPG-CE | 16% | 74% | 10% | 3.67 | 2% | 67% | 31% | 4.71 | 7.12 |
| MADDPG-GE | 27% | 72% | 1% | 3.36 | 2% | 66% | 32% | 4.62 | 6.33 |
| MADDPG-EE | 14% | 81% | 5% | 3.94 | 0% | 85% | 15% | 3.52 | 6.40 |

Table 2: Results for tests depicted in Fig. 4, evaluated during 1000 episodes for each of 7 different random seeds. SR means average success rate, FR means average failure rate, and RATSO is the ratio of average turns to succeed vs. the optimum turns to succeed. 1-SR-FR depicts the ratio of episodes where an agent did not reach any grid cell to terminated the test (either successfully or unsuccessfully) before the trial reached the maximum number of turns allowed ($5w$). A horizontal line means a metric could not be computed.

reward is a valuable overall metric. The low average success rates across all tests show there is significant room to improve in this benchmark. Success rate drops sharply when increasing the grid size, suggesting larger grids impose more difficult training settings. In this analysis, we used models that were trained specifically for each parameter combination.

Results for Oracle were omitted since some tests assumed no knowledge about communication (e.g. even though agents in Figure 4b do not communicate during the test, the test was designed to test if the red agent assumed them to be).

As we emphasized in the main text, many more tests can be proposed. The code base we will release allows for easily adding new tests to the suite.

## A.2 Post-hoc analyses

We developed several post hoc metrics to analyze specific aspects of our models.

- **Unsuccessful recharge base usage rate:** Average times per episode an agent steps on its recharge base without having all the information available (i.e. wrong usage of the recharge base). Note that an agent may step on its base just because it is on the shortest path to another cell. Therefore, a perfect theory of mind agent will likely not have zero on this score; but generally, lower is better. See results in Table 3.

- **Wrong communication piece selection count:** Average times per episode an agent attempted to say an information they currently do not possess. In these cases, no communication happens. Lower is better. See results in Table 4.

- **Useless communication piece selection count:** Average times per episode an agent communicated an information piece that everyone in its hearing range already knew, when having a piece of information that at least one agent in its range did not know. Lower is better. See results in Table 5.

- **Useless movement:** Average times per episode an agent moves away from every agent that does not have the exact same information it has, given that the agent does not currently possess all the information available. This means that the agent is moving away from any possible valuable interaction. Lower is better. See results in Table 6.

All metrics are normalized by number of agents (i.e., they show the score for a single agent). This allows for better comparison between $a = 3$ and $a = 4$ settings.

RMADDPG had the worst scores for unsuccessful recharge base use rate and useless communication piece selection count. RMADDPG scored 43% more than Oracle for unsuccessful base usage on average, and 63% more than Oracle on average for usage of a useless communication piece. The best tailored models (MADDPG-CE and MADDPG-GE) performed similarly to Oracle on average for these two metrics. In contrast, MADDPG-CE and MADDPG-GE performed significantly worse than Oracle for the wrong communication piece selection count (48% and 56% more than Oracle on average). This suggests that all models may be making wrong decisions, but RMADDPG is biased towards communicating redundant information whereas MADDPG-CE and MADDPG-EE tend towards not communicating at all (the true effect of trying to communicate something they are not allowed). Further analysis is needed to truly understand if these apparently wrong behaviors were done in turns where the agent had all the information available to make a better move, or if this is their default when they believe they have nothing of value to communicate. A priori RMADDPG bias seems more principled, but it still showed worse performance overall.

No learned model performed particularly better in the useless movement metric (average differences in performance were less than 10%), suggesting that they perform pointless movements in similar frequencies. It is important not to overinterpret small differences in these metrics. For example, a useless movement may be a signal of emergent communication. Furthermore, an agent may communicate something suboptimal for its immediate reward but this move may not affect its expected reward for the trial.

| agents (a) | | | 3 | | | | | | 4 | | | |
|---|---|---|---|---|---|---|---|---|---|---|---|---|
| grid width (w) | | 6 | | | 12 | | | 6 | | | 12 | |
| info pieces (c) | a | 2a | 3a | a | 2a | 3a | a | 2a | 3a | a | 2a | 3a |
| MADDPG-Oracle | 3.6 | 3.1 | 4.4 | 4.9 | 3.7 | 3.1 | 2.8 | 4.4 | 1.3 | 2.4 | 3.1 | 1.0 |
| MMADPG | 4.8 | **3.5** | 2.9 | 6.7 | 6.3 | 0.4 | 6.6 | 3.5 | 0.8 | 6.5 | 2.7 | 0.6 |
| RMADDPG | 5.9 | 5.0 | 3.2 | 9.9 | 6.3 | 4.7 | 5.4 | 4.8 | 0.8 | 7.3 | 2.6 | 0.5 |
| MADDPG-CE | **4.1** | 5.7 | 3.3 | **4.3** | **2.8** | 0.4 | **3.1** | 3.4 | 1.1 | 4.3 | **1.7** | 0.5 |
| MADDPG-GE | 4.1 | 5.5 | 3.7 | 4.9 | 5.0 | 0.3 | 4.3 | 3.3 | 1.0 | **3.7** | 2.3 | 0.6 |
| MADDPG-EE | 4.8 | 4.1 | **2.4** | 6.2 | 6.4 | **0.3** | 6.6 | **3.0** | **0.7** | 7.0 | 3.5 | **0.4** |

Table 3: Results for **unsuccessful recharge base usage rate**, normalized by agent. Bold lettering represents the best result of a learned imperfect-information model for each setting (lower is better).

| agents (a) | 3 | | | | | | 4 | | | | | |
|---|---|---|---|---|---|---|---|---|---|---|---|---|
| grid width (w) | 6 | | | 12 | | | 6 | | | 12 | | |
| info pieces (c) | a | 2a | 3a | a | 2a | 3a | a | 2a | 3a | a | 2a | 3a |
| MADDPG-Oracle | 3.2 | 3.0 | 3.5 | 10.7 | 10.2 | 5.2 | 2.7 | 5.1 | 2.0 | 7.5 | 6.4 | 3.4 |
| MADDPG | 4.7 | **0.4** | 0.3 | 11.8 | 4.8 | 1.3 | 5.7 | 0.3 | 1.3 | 10.3 | **0.8** | **2.0** |
| RMADDPG | **3.2** | 1.7 | 1.0 | 9.7 | 5.9 | 3.4 | **3.3** | 2.0 | 1.0 | 10.3 | 5.5 | 5.0 |
| MADDPG-CE | 6.2 | 3.8 | 3.5 | 11.4 | 5.3 | 3.8 | 7.0 | 4.2 | 3.7 | 15.5 | 4.7 | 10.9 |
| MADDPG-GE | 6.3 | 3.9 | 3.3 | 11.3 | 6.6 | 3.9 | 7.9 | 4.2 | 3.5 | 14.2 | 7.8 | 11.9 |
| MADDPG-EE | 4.5 | 0.4 | **0.2** | **9.5** | **4.1** | **0.4** | 5.1 | **0.1** | **0.8** | **9.1** | 1.5 | 2.9 |

Table 4: Results for **wrong communication piece selection count**, normalized by agent. Bold lettering represents the best result of a learned imperfect-information model for each setting (lower is better).

| agents (a) | 3 | | | | | | 4 | | | | | |
|---|---|---|---|---|---|---|---|---|---|---|---|---|
| grid width (w) | 6 | | | 12 | | | 6 | | | 12 | | |
| info pieces (c) | a | 2a | 3a | a | 2a | 3a | a | 2a | 3a | a | 2a | 3a |
| MADDPG-Oracle | 8.4 | 13.2 | 14.7 | 20.6 | 26.2 | 19.0 | 10.6 | 17.3 | 12.2 | 24.0 | 17.3 | 12.6 |
| MADDPG | 8.4 | 13.1 | 20.4 | 18.2 | 30.5 | 43.2 | 14.6 | 16.9 | 24.3 | 25.1 | 38.6 | 49.8 |
| RMADDPG | 8.1 | 16.4 | 18.6 | 18.3 | 33.4 | 38.4 | **14.3** | 19.7 | 24.4 | 29.0 | 38.1 | 50.0 |
| MADDPG-CE | 11.8 | 9.5 | **9.8** | 20.7 | 13.3 | **8.6** | 17.4 | 13.2 | **14.8** | 27.2 | **12.1** | 28.4 |
| MADDPG-GE | 12.4 | **9.3** | 10.4 | 19.8 | **12.7** | 9.6 | 16.7 | **11.7** | 16.8 | 24.7 | 17.4 | **27.5** |
| MADDPG-EE | **7.8** | 12.8 | 20.3 | **16.1** | 35.0 | 41.3 | 14.4 | 17.7 | 25.2 | **23.2** | 36.5 | 49.6 |

Table 5: Results for **useless communication piece selection count**, normalized by agent. Bold lettering represents the best result of a learned imperfect-information model for each setting (lower is better).

| agents (a) | 3 | | | | | | 4 | | | | | |
|---|---|---|---|---|---|---|---|---|---|---|---|---|
| grid width (w) | 6 | | | 12 | | | 6 | | | 12 | | |
| info pieces (c) | a | 2a | 3a | a | 2a | 3a | a | 2a | 3a | a | 2a | 3a |
| MADDPG-Oracle | 2.4 | 1.3 | 1.9 | 4.1 | 6.0 | 3.6 | 0.8 | 1.4 | 0.9 | 2.6 | 2.5 | 1.8 |
| MADDPG | **2.1** | 2.2 | **1.4** | 5.0 | 3.9 | **3.1** | 2.1 | 1.2 | 0.7 | 4.4 | 3.4 | 1.4 |
| RMADDPG | 2.8 | 2.5 | 1.6 | **3.8** | 4.2 | 3.7 | 1.9 | 1.5 | **0.4** | **4.1** | 3.3 | **1.2** |
| MADDPG-CE | 2.3 | 2.4 | 1.7 | 5.4 | 2.3 | 3.4 | **1.9** | 0.8 | 1.0 | 4.2 | **1.0** | 3.8 |
| MADDPG-GE | 3.0 | **2.1** | 1.7 | 4.5 | **1.8** | 3.1 | 1.9 | **0.6** | 0.8 | 4.2 | 2.0 | 3.6 |
| MADDPG-EE | 2.4 | 2.2 | 1.7 | 4.2 | 5.0 | 3.4 | 2.2 | 1.3 | 0.5 | 4.6 | 3.5 | 1.9 |

Table 6: Results for **useless movement count**, normalized by agent. Bold lettering represents the best result of a learned imperfect-information model for each setting (lower is better).

## A.3 TRAINING CURVES FOR THREE RANDOM SEEDS COMBINED

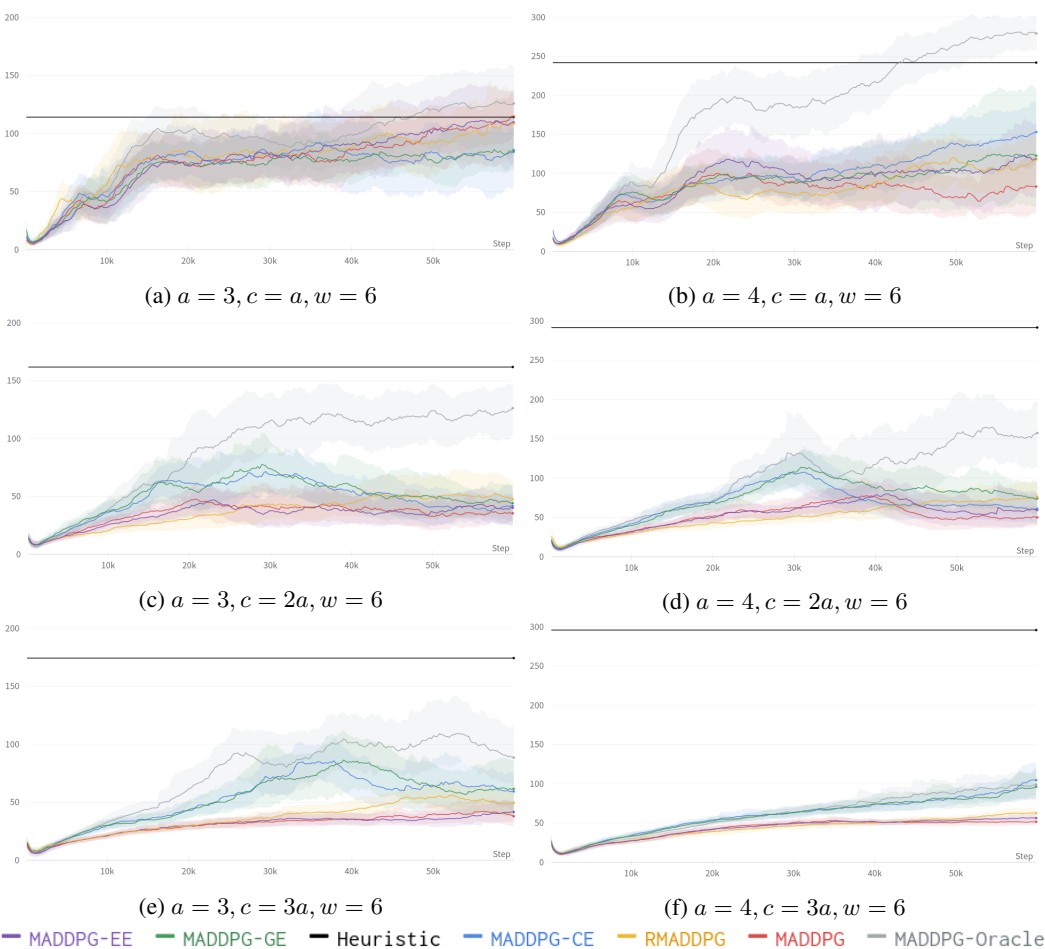

Figure 5: Average episode rewards throughout training for 60000 episodes for all combinations of $a \in \{3, 4\}$, $w = 6$, and $c \in \{a, 2a, 3a\}$.

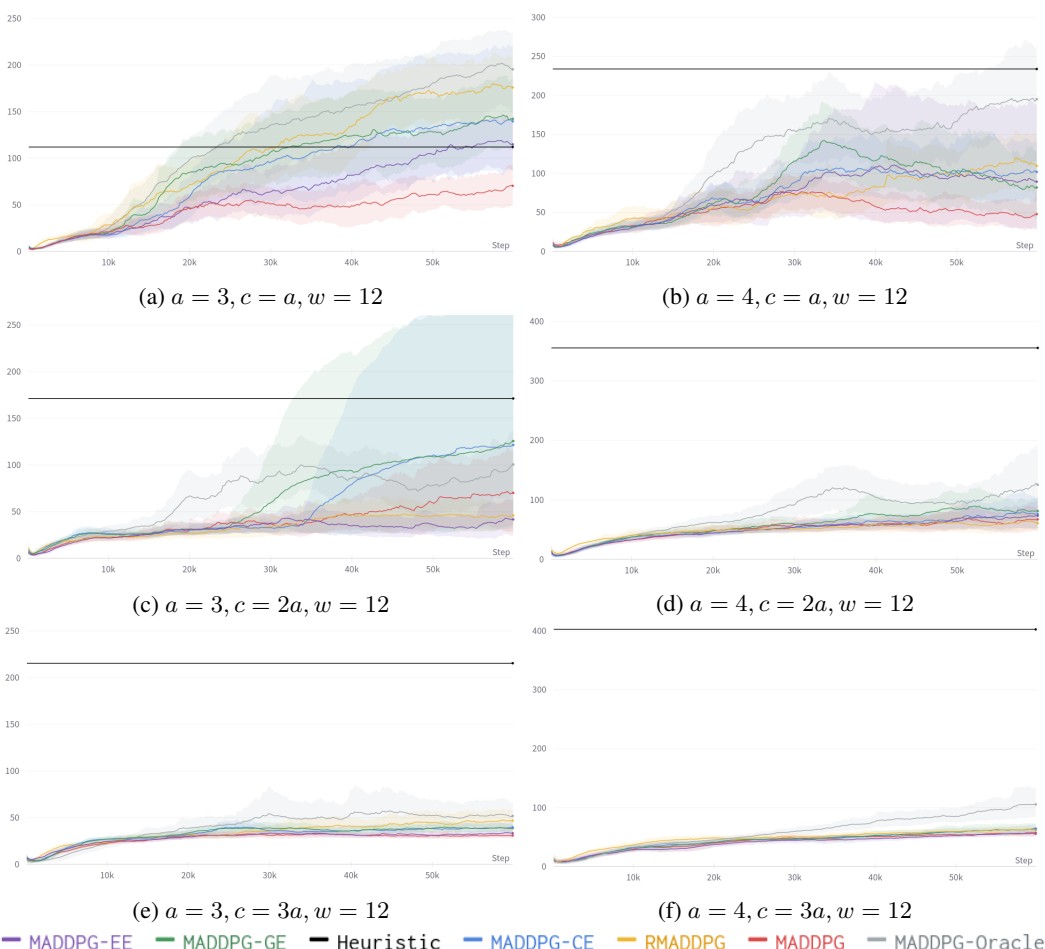

Figure 6: Average episode rewards throughout training for 60000 episodes for all combinations of $a \in \{3, 4\}$, $w = 12$, and $c \in \{a, 2a, 3a\}$.

## A.4 Pseudocode of MADDPG-EE

---

**Algorithm 1** Actor implementation of MADDPG-EE, approximating $K$ to make it differentiable.
Input: *observation*, $A \in \{0,1\}^{c \times a}$, *agent_idx* $\in 0, \ldots a-1$, $F \in \{0,1\}^{c \times a}$, $K \in \{0,1\}^{c \times a}$, $B \in \{0,1\}^a$, $H \in \{0,1\}^{a \times a}$

---

Make agents not be in their own hearing range, to avoid talking to themselves from the previous turn. This would be problematic when using recharge bases.
$$H = H - \mathbb{1}_{a \times a}$$

Compute $S^{[0]}$, all the heard information spoken by *agent_idx*:
$$S^{[0]} = \underset{a \; times}{copy \, A[:, agent\_idx]} \odot \underset{c \; times}{copy \, H[agent\_idx, :]}$$

Compute $S^{[1]}$, all heard information by *agent_idx*, spoken by all agents:
$$S^{[1]} = (A \odot \underset{C \; times}{copy \, H[agent\_idx, :]}) \cdot H$$

Compute $S^{[2]}$, an estimation of information pieces communicated between agents that were out of *agent_idx*'s hearing range:
$$U_j = softmax(f_1(K_{1j}, \ldots, K_{cj}, \{K_{1\ell}, \ldots, K_{c\ell} \text{ for all } \ell \text{ where } H_{j\ell}\})), \text{ with } f_1 \text{ an MLP}$$
$$S^{[2]}_{ij} = 1 - \prod_{\ell, H_{j\ell}=1} 1 - U_\ell \text{ for all } i \in \{0, \ldots, c-1\} \text{ and } j \in \{0, \ldots, a-1\}$$

$$S = S^{[0]} + S^{[1]} + S^{[2]}$$

$$E_i = \mathbb{1}_{sum(K[:,i])=c} \in \{0,1\}^a \quad \text{for all } i \in \{0, \ldots, c-1\}$$

$$K = step\big(F \cdot 100 + K + S - 2 \cdot \underset{c \; times}{copy(B \odot E)}\big)$$

**return** $softmax(f_2([observation \; K])), K \qquad$ where $f_2$ is an MLP

---

## A.5 Experiment Design Decisions and Considerations About Result Presentation

All our experiments are with $h = 1$: only the immediate neighbors of an agent will hear what they communicate.

We tested with $a = 3$ and $a = 4$ and not larger numbers of agents, as the training time increases quadratically with $a$; also, the intrinsic difficulty of larger setup –even with perfect information– would possibly degrade performance to the point of making it impossible to compare models.

Running experiments with the same number of turns for every setting would imply that agents can move less in combinations with larger values of $w$, hence the need of making it proportional to the size of the grid. Since the duration of the experiment is directly proportional to the length of the episodes, we settled on a small multiplier. $5w$ allows agents to move to each edge of the grid and back to the center. A similar decision is required when choosing $c$: having a constant number of information pieces when increasing the number of agents would make the problem easier, as each agent would have fewer options of first-hand information pieces.

We decided to evaluate running additional episodes over the best checkpoints of each model because there was high variance for some runs, and drops in performance after achieving the highest rewards. Those results are the base of the discussion and can be seen in Table 1. Still, we share the training curves so that the reader can observe these behaviors in Fig. 5 and Fig. 6.

We used the same hyperparameters as the ones used in MADDPG, except with a reduced learning rate and tau ($lr = 0.001$ and $\tau = 0.005$). We used the same parameters for all our experiments.

| agents (a) | | | 3 | | | | | | 4 | | | |
|---|---|---|---|---|---|---|---|---|---|---|---|---|
| grid width (w) | | 6 | | | 12 | | | 6 | | | 12 | |
| info pieces (c) | $a$ | $2a$ | $3a$ | $a$ | $2a$ | $3a$ | $a$ | $2a$ | $3a$ | $a$ | $2a$ | $3a$ |
| MADDPG-Oracle | 20 | 16 | 13 | 53 | 35 | 55 | 15 | 20 | 14 | 41 | 25 | 21 |
| MADDPG | 5 | 7 | 2 | 34 | 41 | 2 | 24 | 12 | 2 | 17 | 4 | 3 |
| RMADDPG | 15 | 8 | 8 | 25 | 47 | 18 | 13 | 8 | 6 | 24 | 10 | 3 |
| MADDPG-CE | 15 | 11 | 15 | 32 | 86 | 5 | 20 | 12 | 15 | 29 | 35 | 6 |
| MADDPG-GE | 16 | 9 | 12 | 56 | 70 | 6 | 43 | 16 | 13 | 26 | 51 | 8 |
| MADDPG-EE | 11 | 8 | 6 | 25 | 32 | 2 | 35 | 11 | 3 | 59 | 12 | 4 |

Table 7: Standard deviations of the average episode reward averaged by seed shown in Table 1.

