# OpenReview forum: "Symmetric Machine Theory of Mind"
_ICLR.cc/2022/Conference — ICLR 2022 Submitted_

### Official Review · Reviewer_t9A3 · 2021-11-02

**Correctness:** 4
**Technical Novelty And Significance:** 2
**Empirical Novelty And Significance:** 4
**Recommendation:** 6
**Confidence:** 4

**Main Review:**

Strengths:

The topic covered in the paper is extremely timely: ToM, and MARL in general (specifically centralized training with decentralized execution (CTDE) MARL such as MADDPG), are topics receiving a great deal of interest, with many workshops and conference publications related to this work.

The paper makes a compelling case for SymmToM with both the motivating discussion and the results. The definitions of "Theory-of-Mind Agents" and "Symmetric Theory-of-Mind environments," while perhaps having some debatable components (e.g. is it important that each agent _always_ get some information about every other agent, as seems to be implied?), are reasonable and in my mind cover many interesting cases. The definition of SymmToM follows cleanly from this. It is a very simple environment that seems to contain minimal aspects critical for it to count as a Symmetric Theory-of-Mind environment. Finally, the results make a compelling case that this is a useful benchmark. In particular, by tuning the number of agents and info pieces, we easily reach a regime in which Heuristic substantially dominates several reasonable baselines.

These baselines are extremely hand-crafted for the environment, which would detract if the point of the paper were to demonstrate their utility. But that is not the point: the point is that (1) there is a substantial delta between these and Heuristic, and (2) that they are so hand-crafted that I cannot think of a way in which more generic baselines with more sophisticated methods could reasonably be expected to do better. This means that there is demonstrably considerable room to improve.

The paper is well-written and easy to read, with useful figures.

Weaknesses:

It is important to think critically about whether SymmToM provides a robust test for ToM. Intuitively, several things are challenging about this environment: the need to estimate the belief/knowledge states of others, coordination (generating mutually beneficial behaviors between agents -- as Heuristic does in a very nontrivial way), and long-range planning (collecting all these info pieces before going to the recharge bases.

With regard to the need to estimate the belief/knowledge states of others, one critique is that this is a drastically simplified parametrization of belief/knowledge. That is, belief/knowledge is not about embodied/spacial/actionable knowledge about the environment, but rather it is abstracted away into discrete info pieces and thus lacks real-world nuance (e.g. one might try to estimate the location of something and know roughly where it is, and roughly where others think it is, but not have an exact sense of either; or one might be attempting to estimate others' goals). It is thus a bit unclear just how much this is capturing in our intuitive, psychological notion of theory of mind.

Coordination is a massive challenge in CTDE MARL. In a sense, the Heuristic baseline largely solves a coordination problem: with this sort of simple behavior, ToM estimation and long-term planning is seems relatively easy. What might be difficult is not necessarily ToM estimation but rather simply achieving this level of coordination between agents. Failing massive amounts of coordination, planning in this environment seems to be very challenging as well, easily being difficult for long-range planning algorithms simply because of the number of subgoals one needs to achieve in order to get the recharge reward.

A few things (not necessarily in the scope of this paper) could help clarify this.

(1) Perhaps easiest, a direct analysis of the baselines' capacity to estimate K, as well as a ceiling for K estimation, would be useful. If it turns out that the models are already estimating K well, then this is perhaps more a benchmark for other challenges, not ToM.
(2) It would be helpful to see what humans do on this task. One would be to see how often humans, playing together for a while, converge to simple, effective cooperative behaviors such as Heuristic.
(3) The other would be to see how well a human does when the other agents are not particularly coordinated. One possibility is that the amount of things needed to be kept track of in the higher a, c environments simply make for not particularly human-doable tasks.

**Summary Of The Paper:**

This paper introduces SymmToM, an environment aimed at benchmarking agents' Theory of Mind (ToM) capabilities. In motivating the construction of this environment, it outlines criteria.

- It first defines ToM agents in terms of knowledge/prediction of each agent's own hidden state,

- It then defines "Symmetric Theory of Mind" environments as having certain properties: symmetric action space, imperfect information, observation of others, and information-seeking behavior (really, the task-relevance of it). This is motivated by attempts to move away from more traditional ToM situations, e.g. where one agent is a passive observer or agents have one of several different designated roles.

With this, it defines the environment, a simple grid environment with several modifiable parameters.

It then defines several reasonable baselines: an oracle (that knows info states of all agents but must learn what to do with this via MADDPG -- so, not necessarily an upper bound), an heuristic (consisting of a simple coordinated strategy between agents that is also not an upper bound, but represents decent coordinative behavior), and several hand-crafted agents that explicitly track knowledge estimates (these have a great deal of domain-specific architecture).

It then tests each of these baselines on several versions of the environment (varying the aforementioned parameters). In all but the simplest versions, the heuristic method dwarfs the performance of everything else (including the oracle), thereby demonstrating room to improve.

**Summary Of The Review:**

In sum, the paper puts forth a compelling benchmark: one that is simple and well-motivated but appears to demonstrate a considerable delta between an heuristic "upper bound" (not necessarily actually the upper bound) and reasonable baselines. I feel that these merits outweigh the critique that this might not be isolating critical aspects of ToM capacity in people. As such, I think it would be worthwhile to get this to the broader community.

---

> ### Author Response · Authors · 2021-11-23
> **Response to reviewer t9A3**
>
> We thank the reviewer for their thoughtful feedback! We are encouraged that they found our paper timely and well-written, and SymmToM to be a compelling benchmark. We reply to their comments below.
>
> > “It is important to think critically about whether SymmToM provides a robust test for ToM. Intuitively, several things are challenging about this environment (...)”
>
> We agree with the reviewer that SymmToM requires multiple behaviors to be solved: modeling knowledge about others (the ToM goal itself), as well as coordination and long-range planning. We are convinced that this combination of challenges is necessary to truly test multi-agent theory of mind. Coordination is required for most cooperative multi-agent ToM scenarios, since multiple parties benefit from a single interaction, and there is pressure to earn reward as fast as possible. Long-range planning will also be required since we need belief states to evolve over time: to have different perspectives/beliefs about the same situation, we need a complex scenario to evolve over the course of several turns; this is true even without involving recharge bases.
>
> > “Is it important that each agent always get some information about every other agent, as seems to be implied?”
>
> We clarified in the manuscript that we do not believe Symmetric Theory of Mind requires each agent to get partial information about all agents in *every* turn, but rather that this information cannot always be null, since otherwise an agent would be invisible to another. In a scenario with restricted vision, for example, an agent may not get information about a specific agent that is too far apart. Symmetry also implies that if A observes some particular aspect about B, the reverse should also be true if the roles were to be reversed. We would like to thank the reviewer for pointing out this unclear phrasing.
>
> > SymmToM is a drastically simplified parametrization of belief/knowledge
>
> We agree SymmToM does not reflect all nuances of the real-world! We firmly believe that by showing that even this simplification cannot be solved with our current models, we can better direct immediate next steps in machine theory of mind modeling research.
>
> > Coordination vs. ToM vs. Long term planning measurement and importance for solving the task. Suggestions (measuring K and human comparison).
>
> We want to thank the reviewer for the useful suggestions on this topic. We have included the results of tests discussed in Section 7.3 in the appendix to more specifically measure different aspects of theory of mind, that may be more difficult to infer solely from a global reward metric. We will add more discussion about these new results in the main text for the camera ready version.
>
> We agree that it would be interesting to test humans on this task, and we also hypothesize that they may converge to a suboptimal policy --like the heuristic-- to compensate for the constrained amount of memory humans have, and the impossibility to methodically update and estimate knowledge about other agents. These should not be limiting factors for agents and therefore we would expect even better performance in agent-agent interactions. Keeping these limitations in mind, if we experiment with human-agent interaction we will make sure to maintain parameters such as $a$ and $c$ as reduced as possible.
>
>
>
> We would like to thank the reviewer again for their insightful feedback!

---

> > ### Comment · Reviewer_t9A3 · 2021-11-23
> > **Thanks!**
> >
> > Thanks for your responses! As I gave a higher score than others (and their critiques raised additional concerns), I will leave my score as-is for now and modify upon seeing their responses if I feel it is appropriate.

---

### Official Review · Reviewer_n7Fh · 2021-11-02

**Correctness:** 3
**Technical Novelty And Significance:** 3
**Empirical Novelty And Significance:** 3
**Recommendation:** 3
**Confidence:** 4

**Main Review:**

This work considers a broader range of tasks by removing a pre-defined role of the agent. From the symmetric setting, every agent can play all roles without being biased towards a specific role.

Even though the paper has merit, the reviewer has the following concerns.
1)  The paper adopts the partially observable Markov decision process. But in section 7.1, the observation space is defined by a set that consists of the position of **all** agents, **all** recharge bases, **every** agent’s first-hand information, etc. It does not seem to be a “partial” observable setting. In addition, the assumption that the agent has full vision seems not realistic in a large grid world. How can every agent know the initial knowledge of the other agents?

2) Recharge bases seem to be important in the proposed environment. However, the process that happened in the recharge bases is not fully explained. Why all the information gathered are removed? It would be better the information pieces are removed depending on the importance, or time.

3) The paper must include an exact definition of reward. The reward function is not provided in the paper.

4) The experimental results in Table 1 were written with only 3 runs, which is a very insufficient number of trials to explain the claim. In addition, it is not a good method to judge the performance of a specific algorithm through the average of the results performed under different experimental conditions.

5) The values in Table 1 do not have enough explanations. Does it represent a ‘per agent’ reward?

6) The paper should be reorganized. For example, Fig. 2 does not have specific captions(a/b/c), so it is hard to follow. In addition, Fig. 3 has 4 images with different sizes and #1(line 3 in the caption) is not explained.

7) No parameter tuning and hyperparameter setting methods for experiments

**Summary Of The Paper:**

This paper suggests a fully symmetric multi-agent environment based on the Theory of Mind, SymmToM. In the SymmTom, agents can remember its information, infer behaviors of other agents, and estimate the probability of others. Interactions between agents are modeled by a partially observable Markov decision process. Each agent decides an action(move, speak) upon its state which consists of partial observations. The authors admit that the proposed SymmTom cannot be completely solved. They adopt MADDPG for performance evaluations.

**Summary Of The Review:**

The multi-agent reinforcement learning with the Theory of Mind is an interesting topic to be discussed. The proposed SymmTom aims at the researchers in the Theory of Mind and MARL field. But overall problem setups and experimental results are not described enough.

---

> ### Author Response · Authors · 2021-11-23
> **Response to reviewer n7Fh**
>
> We thank the reviewer for their detailed feedback, and we answer below.
>
> > 3 runs is a very insufficient number of trials to explain the claim.
>
> We recomputed all the results with 7 random seeds for each combination, and the results are consistent with what we had shown prior. Running with more seeds made it even more clear that RMADDPG did not perform as well as MADDPG-*E models, and that the gap between these and the heuristic&oracle is significant. We have included stdev metrics in the appendix, and will work on a more legible approach for the camera ready. We would also like to point out that we are showing 72 different combinations of parameters, and therefore each additional run requires significant computational power. For comparison, the original MADDPG paper showed the average of 3 random seeds for some environments, and 10 for others. We have 3 more runs in progress that will be added for the camera ready.
>
> > The paper must include an exact definition of reward.
>
> We have added an equation defining reward in Section 4.
>
> > "The authors admit that the proposed SymmToM cannot be completely solved"
>
> This is exactly what we want to emphasize. As reviewer t9A3 highlights, the SymmToM is very simple and even the most tailored baselines fail to solve it, proving there is significant room for future modeling research, and showing the usefulness of SymmToM as a benchmark.
>
> > Concerns about SymmToM’s definition: 1. It does not seem to be a “partial” observable setting. 2. Agent with full vision do not seem realistic in a large grid world. 3. Why are recharge bases defined like this instead of knowledge being removed proportional to time or importance? 4. How can every agent know the initial knowledge of the other agents?
>
> 1. SymmToM is a partially observable setting. Partial observability comes from being unable to hear interactions outside the agent’s specific hearing range (in section 4 we say: “Messages sent by agents outside of the hearing range will not be heard”).
> 2. SymmToM is a simplified scenario to test if models can show theory of mind behavior in symmetric settings, and we found that handcrafted models fail even without introducing additional rules. The code base that we will release allows for exploring other observation space limitations including restricted vision (as mentioned in the conclussion), but we focused on showing experiments on the simplest scenario possible where models fail.
> 3. In a similar vein, we do not claim recharge bases to be the most realistic way of forgetting information, but it is an explicit way for other agents to infer information loss, and for the researchers to measure zeroth-order theory of mind.
> 4. Having public initial knowledge is also a simplification to test our models in a more manageable setup, where we show that current models cannot perform well. One intuitive metaphor for public initial knowledge is to think that agents hold their first-hand information in numbered envelopes: everyone can see the information piece numbers each person holds initially, but they do not immediately learn the information since it is hidden inside the envelope. The only way of learning information we do not have in our envelopes is through communication. Having private initial knowledge is a more complex and very interesting future direction, as mentioned in section 7.1.
>
> > It is not a good method to judge the performance of a specific algorithm through the average of the results performed under different experimental conditions.
>
> We have removed the column that averages individual scores in different settings. We would like to clarify that the results discussion was not based on that column. For example, when in section 7.2 we report “A is 125% better than B on average”, we are basing our claim on the average of the ratios for each {a,c,w} combination ( \avg_{a,c,w} score{A_{a,c,w}} / score{B_{a,c,w}} ).
>
> >  Does Table 1 represent a ‘per agent’ reward?
>
> Yes! The original caption stated “Values shown are individual rewards”, but we have rephrased it to emphasize it represents a per agent reward.
>
> > hyperparameter setting methods?
>
> We based our hyperparameters on the ones used in MADDPG, but reduced the learning rate and tau to be able to reproduce their results (lr=0.001; tau=0.005). We used the same parameters for all our experiments. We would like to emphasize that the point we are trying to convey is that all the learned models are significantly below the heuristic&oracle (for as much as an order of magnitude).
>
> > Fig. 2 does not have specific captions(a/b/c), so it is hard to follow. In addition, Fig. 3 has 4 images with different sizes and #1 (line 3 in the caption) is not explained.
>
> We have fixed these points in the manuscript, thanks for pointing them out! Images have different sizes to allow grid cells to be of a legible size for all examples without taking more space.
>
> We thank the reviewer again for their feedback and we hope to have addressed their concerns.

---

> > ### Comment · Reviewer_n7Fh · 2021-11-29
> > **regarding authors' response**
> >
> > I have read the response that the authors provided. They addressed a part of my concerns and have been improving the manuscript. Even though their efforts, I still think the current partial observation setting has too many unrealistic assumptions. I would stick to my current score.

---

### Official Review · Reviewer_sDZt · 2021-11-02

**Correctness:** 3
**Technical Novelty And Significance:** 2
**Empirical Novelty And Significance:** 2
**Recommendation:** 5
**Confidence:** 4

**Main Review:**

=====Strengths=====

1. Testing RL agents or any kinds of machine agents' ToM ability is an important and yet understudied problem. When we evaluate the success of multi-agent policies, it is important to test the true social intelligence that comes with the policies. Naturally, that includes ToM reasoning. So this is certainly a welcomed contribution in the area of multi-agent RL in my opinion.

2. The explicit modeling of agents' knowledge is an interesting extension to MADDPG. It makes sense to use domain knowledge in some cases to improve the RL training if there is a discussion on the limit or possible improvement.

3. The two kinds of tests proposed in the discussion are very interesting and are to some extent the most important aspect of this study -- evaluating and analyzing the true ToM ability of agents trained in SymmToM that goes beyond reporting a single reward value.

=====Weaknesses=====

1. The literature is insufficient. There has been a rich history of computational modeling of ToM (e.g., [1,2,3,4]). There have also been multiagent communication and cooperation tasks / environments proposed before (e.g., the particle environment proposed in the MADDPG work, and [5,6]). There should be a more thorough discussion and comparison.

2. It is also unclear to me why it is necessary to have a symmetric setting. This setting is emphasized in the title and in the main text, but I have not seen the motivation for it.

3.  I do not see a systematic and quantitative evaluation of the two kinds of tests proposed in the discussion section. Am I missing something here? Is it more of a proposal than a completed evaluation?

4. There should be a discussion on how general and scalable the explicit modeling of knowledge is as an extension to MADDPG or similar multi-agent RL approaches.

References:

[1]  T. D. Ullman, C. L. Baker, O. Macindoe, O. Evans, N. D. Goodman and J. B. Tenenbaum (2010), Help or hinder: Bayesian models of social goal inference. In NeurIPS.

[2]  Netanyahu, A., Shu, T., Katz, B., Barbu, A., & Tenenbaum, J. B. (2021). PHASE: PHysically-grounded Abstract Social Events for Machine Social Perception. In AAAI.

[3] Zhu, H., Neubig, G., & Bisk, Y. (2021). Few-shot language coordination by modeling theory of mind. In ICML.

[4] Shu, T., Bhandwaldar, A., Gan, C., Smith, K. A., Liu, S., Gutfreund, D., ... & Ullman, T. D. (2021). AGENT: A Benchmark for Core Psychological Reasoning. In ICML.

[5] Das, A., Gervet, T., Romoff, J., Batra, D., Parikh, D., Rabbat, M., & Pineau, J. (2019). Tarmac: Targeted multi-agent communication. In ICML.

[6] Jain, U., Weihs, L., Kolve, E., Rastegari, M., Lazebnik, S., Farhadi, A., ... & Kembhavi, A. (2019). Two body problem: Collaborative visual task completion. In CVPR.


**Summary Of The Paper:**

This paper presents a multi-agent environment and task, termed SymmToM, for analyzing machine theory of mind emerged from multi-agent RL training. In SymmToM, agents can see and act in a 2D grid world as well as share information about the world state through communication. Crucially, all agents have the same physical characteristics, hence, "symmetric." In this task, agents gain rewards by hearing or sharing novel knowledge of the 2D grid world, so that the optimal policy would be cooperatively seeking and sharing information among the agents. The proposed approach extends MADDPG to explicitly model the knowledge of each agent and how it would be affected by the shared information. The experimental results show that this extension improves over the generic RNN based extension to MADDPG (i.e., RMADDPG), but is still not performing as well as the simple heuristics-based baseline.

**Summary Of The Review:**

I think this paper has a lot of potentials but is incomplete in its current form due to 1) a lack of interactive review, 2) insufficient evaluation (missing systematic evaluation of the proposed tests), and 3) a lack of discussion of how scalable and generalizable SymmToM and the approach are. Please see specific concerns in my main review.

---

> ### Author Response · Authors · 2021-11-23
> **Response to reviewer sDZt**
>
> We thank the reviewer for their feedback and very relevant pointers. We are working on including a more thorough discussion about related work in the final version of the manuscript, starting from including [1, 2, 4, 5, 6] ([3] had already been included). We answer all comments below.
>
> > Quantitative evaluation of the two kinds of tests proposed?
>
> We have added results of both types of analysis in the appendix, and will include more discussion of them for the camera ready version. We also expanded the number of runs throughout the paper (now we have 7 random seeds), with consistent results. We would like to emphasize that we believe that reward is the best metric to globally analyze performance. The discussion in Section 7.3 aimed to show that there are mainly two different ways to locally measure different levels/aspects of theory of mind in specific setups, that both could be useful to measure particular phenomena, but none generalize to a global metric as well as using the reward.
>
> >  How general and scalable is the explicit modeling of knowledge?
>
> We would like to clarify that the main goal of analyzing models with explicit modeling of knowledge is not to propose them as the best solution for all tasks --although they can be useful for some, like SymmToM--. In contrast, we aim to show that even models tailored specifically to our task fail to solve this simple scenario (as reviewer t9A3 insightfully points out), showing we need more research in this area and proposing a useful benchmark to tackle it.
>
> > “It is also unclear to me why it is necessary to have a symmetric setting.”
>
> We believe that symmetric settings are crucial to study in theory of mind, and present additional challenges. So far, research has focused mainly on speaker-listener and passive observer scenarios. In speaker-listener settings, the speaker optimizes to the invariant that only them will be able to communicate information but not move/perform, and the listener optimizes their behavior to be an executor only. A single agent never learns both abilities, and more importantly, never learns to decide between communication and movement/execution. One example of this decision-making in SymmToM is when an agent needs to decide whether to move to their recharge base or to keep communicating their current knowledge.
>
> Additionally, in passive observer scenarios the observer learns to judge a scenario where fully-trained observers interact with each other, but this observer cannot modify or intervene in any way. Besides the question of this passive-predicting knowledge will translate to a scenario where the agent has to actually interact with others, our setup comes with additional learning complexities. For example, during training agents often interact with imperfect players, possibly leading to more flexible agents. We did not test this hypothesis in the present work, but believe it would be an interesting future direction.
>
> We thank the reviewer again for their feedback and we hope to have addressed their concerns.

---

> > ### Comment · Reviewer_sDZt · 2021-11-30
> > **Thank you for the responses**
> >
> > Thanks for your responses. While it is great to see more evaluation results (I have increased my rating because of this), I do not agree that you would need a symmetric setting to go beyond speaker-listener and passive observer scenarios. For instance, in the MADDPG paper and in the suggested reference [4], there are settings where agents need to infer the goals of others while moving in an environment and/or communicating with one another. I think the paper would greatly benefit from a major revision.

---

### Official Review · Reviewer_VGt1 · 2021-11-02

**Correctness:** 4
**Technical Novelty And Significance:** 2
**Empirical Novelty And Significance:** 2
**Recommendation:** 3
**Confidence:** 4

**Main Review:**

The problem of imbuing systems with a theory of mind is an interesting one and one that is getting attention in multiple areas in AI. Many works argue for not only modeling the beliefs and knowledge of other agents but also argue that in many scenarios where the agent is interacting with humans, it would need to explicitly take into account what the human expects from the agents (the authors could check works done in explanation in deterministic planning setting for some related work [1]). While the general direction is interesting, I don’t think the paper is ready for publication as the current paper doesn’t make a convincing argument for the novelty or significance of the current results.

Unfortunately, the current version of the paper doesn’t contain a related work section. This is a particularly glaring omission for a paper that is looking at as widely studied a problem as modeling of other agents for multi-agent decision-making settings. Apart from works that frame such modeling of other agents as building theories of mind, which generally happens when there is a human in the setting or if the work is inspired by the psychological phenomena, many (if not most) multi-agent planning itself allows some such form of modeling (even ignoring all the various game-theoretic formulations). Not to mention, there is a pretty mature and well-investigated sub-area of reasoning and planning called epistemic reasoning that has looked at the problem of modeling beliefs and knowledge of other agents for a long time (authors can check [2] for a useful starting point).

A related work section helps the reader compare the contributions of a specific paper against all the previous works that have been done in related areas. For example, in the paper, there was no discussion on how the current problem being proposed compares to multi-agent planning frameworks like DEC-POMDPs [3] or I-POMDPS [4]. Particularly since one of the baselines used in the paper is specifically a solution method for DEC-POMDPs. To the best of my understanding, the specific scenario discussed in the paper seems to be a special case of a DEC-MDP if you allow the facts that the agent collects to be modeled as part of the state. If that is not the case, then the paper should explain why it is not so.

This brings me to the specific problem being studied in the paper. Why is this problem of particular interest? Does it correspond to any particular practical problems? The methods introduced here leverage specifics of the problem and as such, the usefulness of methods are also limited by the utility of the setting. Additionally, the paper doesn’t provide any complexity analysis of this particular problem type or an optimal solution. It was also not clear to me why forgetting information is particularly important for information-seeking behavior. If a given agent has partial observability, the task is the infinite horizon and its utility depends on the other agent’s state and actions, you could always create a setting where the optimal strategy for the agent involves seeking out the information about other agents' state.

Also, the beginning of the paper makes some connections to human-agent interaction, if this is one of the problems the authors are interested in, then they are generally incompatible with any centralized planning mechanisms. Since in most cases you can’t plan for the human. It might make more sense for the authors to base their framework on decentralized planning frameworks like I-POMDPs or some game-theoretic formulation.

[1] Sreedharan, Sarath, Tathagata Chakraborti, and Subbarao Kambhampati. "Foundations of explanations as model reconciliation." Artificial Intelligence 301 (2021): 103558.

[2] Fagin, Ronald, et al. Reasoning about knowledge. MIT press, 2003.

[3] Oliehoek, Frans A., and Christopher Amato. A concise introduction to decentralized POMDPs. Springer, 2016.

[4] Gmytrasiewicz, Piotr J., and Prashant Doshi. "A framework for sequential planning in multi-agent settings." Journal of Artificial Intelligence Research 24 (2005): 49-79.

**Summary Of The Paper:**

The paper focuses on establishing the framework of symmetric TOM. Paper identifies such decision-making settings as being characterized by four different properties, namely, symmetric action space, imperfect information, observation of others, and information-seeking behavior. The paper then introduces a sample task that instantiates the symmetric TOM setting and then formulates some solution approaches for the setting that builds on an earlier multi-agent actor-critic framework, with an unmodified version being the baseline. The proposed method, particularly the GreedyEncounter variant, seems to perform better than some of the other baselines, but not as well as the heuristic version.

**Summary Of The Review:**

As of right now I won't argue for accepting this paper, as the paper doesn’t make a convincing argument for the novelty or significance of the current results.

---

> ### Author Response · Authors · 2021-11-23
> **Response to reviewer VGt1**
>
> We thank the reviewer for their feedback, and we reply to their comments below.
>
> > Current version of the paper doesn’t contain a related work section
>
> We are working on creating a related work section for the camera ready version. We have started with the relevant pointers the reviewer mentioned. The current version only discusses prior work in the introduction due to space constraints we are actively working to solve.
>
> > the specific scenario discussed in the paper seems to be a special case of a DEC-MDP if you allow the facts that the agent collects to be modeled as part of the state
>
> SymmToM is a POMDP even if we include zeroth-order information as part of the state. Partial observability comes from being unable to hear interactions outside the agent’s specific hearing range (please refer to section 4, where we say: “Messages sent by agents outside of the hearing range will not be heard”). Even if an agent can see two agents being in range of each other, it cannot hear what they communicated and has to estimate it based on the knowledge it has from each agent (first-order ToM), the knowledge it has about each agent’s knowledge about other agents (second-order ToM), and so on. We added additional emphasis to this point in Section 7.1.
>
> > “The methods introduced leverage specifics of the problem and as such, the usefulness of methods are also limited by the utility of the setting”
>
> We are not claiming these task-tailored models to be our main contribution. As reviewer t9A3 insightfully points out, the main goal of having models tailored specifically to our task is not just to claim an improvement over the baselines, but rather to show that even handcrafted models fail to solve this simple scenario. This implies that there is significant modeling research to be done before even attempting to solve more realistic multi-agent RL problems requiring theory of mind, such as embodied agents or agents communicating in more complex language.
>
> > “No convincing argument of significance and novelty of the current results” / “Why is this problem of particular interest?”
>
> We have re-run all the experiments with 7 random seeds, showing consistent results with the ones we had before: we proved that even tailored models fail to solve SymmToM (even though they do make improvements over MADDPG and RMADDPG baselines). ​​This makes SymmToM a compelling benchmark since it is a simplified knowledge theory of mind task that remains unsolved, and the only one to our knowledge that does not follow either the passive observer or the speaker-listener paradigm.
>
> We also added results for the analyses discussed in Section 7.3 in the appendix for this rebuttal, to have additional ways of analyzing these results beyond reward.
>
> > “It was also not clear to me why forgetting information is particularly important for information-seeking behavior”.
>
> Assuming we are designing an environment where an agent’s reward is tied to the amount of information it has; that there is a finite amount of information to be gathered, a finite number of agents, a finite environment, and no way of forgetting information; eventually all the information in the environment will be gathered. Even with infinite horizon, it is undesirable to have agents with no possibility of gaining any reward until the end of the episode.
>
> > “the paper makes some connections to human-agent interaction, if this is one of the problems the authors are interested in, then they are generally incompatible with any centralized planning mechanisms”
>
> In the present work we focus solely on agent-agent interactions, but we believe that human-agent interaction is a relevant area. Further research is needed to assess human-agent interaction regarding theory of mind, but a priori we do not find MADDPG agents to be incompatible with human interaction: while it is true that MADDPG has centralized training, its execution is decentralized.
>
> We would like to thank the reviewer for their feedback and we hope to have addressed their concerns.

---

> > ### Comment · Reviewer_VGt1 · 2021-11-28
> > **Re: Author Response**
> >
> > I thank the authors for their response. Below are some of my thoughts on certain points raised by the authors.
> >
> > Dec-MDP: Apologies for not giving a useful reference to this formalism (it's part of the textbook I referred to, but maybe it is hard to find in that book). Dec-MDPs are *not* MDPs, rather they are special cases of DEC-POMDPs where the total of all the observations received by the agents defines the complete state. Which to the best of my knowledge is what's happening here. This doesn't mean each agent on its own has the full knowledge. You can refer to [1] to get a sense of the models and their relation to DEC-POMDP. Also, I am also unsure about the claim being made here that you have a POMDP. Since you have a case with centralized planning and decentralized execution, it should at the very least be a DEC-POMDP.
> >
> > Forgetting Knowledge: My point was just that a state-based representation as employed by most existing multi-agent decision-making systems can easily capture this behavior. I didn't see a particularly compelling reason to separate it out. Does it define a particular sub-class of problems that is of practical importance? And does this specific modeling provide any additional computational advantage?
> >
> > [1]  Bernstein, Daniel S., et al. "The complexity of decentralized control of Markov decision processes." Mathematics of operations research 27.4 (2002): 819-840.

---

### Official Review · Reviewer_cEGn · 2021-11-07

**Correctness:** 2
**Technical Novelty And Significance:** 2
**Empirical Novelty And Significance:** 2
**Recommendation:** 5
**Confidence:** 2

**Main Review:**


The paper introduces the concept of theory of mind well and does a good job at motivating the papers direction.

The proposed environment is well motivate dwith respect to the target of theory of mind. Although, I do think it's important to make sure to qualify the term theory of mind a bit more - as the original concept in psychology does include additional knowledge such as beliefs, desires, intentions and emotions. Most of which are not captured in this paper at all. I'd also really suggest to stay away from misleading terms like "awareness". E.g. I have trouble with the definition or model non-ToM. Theory of mind is about other agents not oneself. It's also not true that the agent is not "aware". It's just that he doesn't have the knowledge in the first place.

Multi-agent settings typically suffer strongly from varied outcomes. It would be important to have more random seeds and also report measures of dispersion of run outcomes. Otherwise, I am not sure these results actually hold true in rigorous statistical testing.

The discussion of potential tests for ToM in 7.3 is interesting. It's not clear though why you don't just apply some of these, in order, to get good statistical data about the performance of trained agents besides accumulated reward.

Why did you choose MADDPG and RMADDPG?

**Summary Of The Paper:**

The paper proposes a multi-agent environment to develop and test theory of mind capabilities for agents. The environment is one where all agents speak, listen to and move around. Agents share information and the goal for each agent is to get all the information and then release. The paper suggests some architecture that avchieve some improvement over agents without theory of mind capabilties. Ultimately though, the task is unsolved and posed as a research question


**Summary Of The Review:**

The paper presents an interesting new challenge that could give rise to a number of new research ideas. The challenge is well motivated from the literature

Authors explore the new task with existing methods with some obvious expansions.

The proposed baselines should be explored more and specific tests outside of reward should be part of actualy analysis

PRO
* paper is written well
* the challenge is well described
* paper presents some sensible baselines

CONS
* no technical contribution
* potentially shaky results (not clear if these results are significant)
* evaluation relies on reward only. other evaluations are discussed but not really explored in detail

---

> ### Author Response · Authors · 2021-11-23
> **Response to reviewer cEGn**
>
> We thank the reviewer for their constructive feedback, and we are encouraged that they found the paper was well written and that they found our baselines to be sensible. We answer their comments below.
>
> > "It would be important to have more random seeds and also report measures of dispersion of run outcomes."
>
> We recomputed all the results with 7 random seeds for each parameter combination, and the results are consistent with what we had shown prior. Running with more seeds made it even more clear that RMADDPG did not perform as well as the MADDPG-*E models, and that the gap between these and the heuristic & oracle is significant. We have included stdev metrics in the appendix, and will work on a more legible approach for the camera ready version. We would also like to point out that we are showing 72 different combinations of parameters, and therefore each additional run requires significant computational power. For comparison, the original MADDPG paper showed the average of 3 random seeds for some environments, and 10 random seeds for others. We have 3 more runs that did not finish in time for including them now, but will be added for the camera ready version.
>
> > Other evaluations besides reward are discussed but not explored in detail
>
> We have added results for both types of analysis (discussed in section 7.3) in the appendix, and will add more discussion about them in the main text for the camera ready version. We would like to emphasize that we believe reward is the best metric to globally analyze performance. The discussion in Section 7.3 aimed to show that there are mainly two different ways to locally measure different levels/aspects of theory of mind in specific setups, that both could be useful to measure particular phenomena, but none generalize to a global metric as well as using the reward.
>
> > “The paper suggests some architecture that achieves some improvement over agents (...)”
>
> We would like to clarify that the main goal of showing models tailored specifically to our task is not to just claim an improvement over the baselines, but rather to show that even models handcrafted for the task fail to solve this simple scenario (as reviewer t9A3 insightfully points out).
>
> > Why MADDPG?
>
> We use MADDPG since it is a well known model in multi agent RL, that allows for decentralized execution. Moreover, MADDPG also generalizes well to competitive scenarios, which we have not explored in this work but could be a potentially interesting future direction.
>
> > Use of term “zeroth-order ToM”
>
> We use the term “zeroth-order theory of mind” following literature from psychology and cognitive science [1, 2], and we made sure to include it in our discussion since it is a crucial ability to perform well in SymmToM, as is the case of most scenarios where the agent is an active player in the environment.
>
> [1] Flobbe, L., Verbrugge, R., Hendriks, P., & Krämer, I. (2008). Children’s application of theory of mind in reasoning and language. Journal of Logic, Language and Information, 17(4), 417-442.
> [2] Hedden, T., & Zhang, J. (2002). What do you think I think you think?: Strategic reasoning in matrix games. Cognition, 85(1), 1-36.
>
> > Theory of mind involves more than just reasoning about other’s knowledge
>
> Thanks for pointing out this was unclear in the text! We have added a footnote to clarify the subset of theory of mind we will be referring to (following what [3] does for their benchmark about understanding goals of others).
>
> [3] Kanishk Gandhi, Gala Stojnic, Brenden M. Lake, and Moira R. Dillon.  Baby intuitions benchmark (bib): Discerning the goals, preferences, and actions of others. CoRR, abs/2102.11938, 2021.
>
> > Misuse of term “awareness”
>
> Thanks for pointing this out, we updated the manuscript to avoid using it. it was not our intention to make a strong psychological claim by using the term “awareness”.
>
> We would like to thank the reviewer again for their useful feedback!

---

### Author Response · Authors · 2021-11-23
**Summary of most important changes**

We thank all the reviewers for their feedback! We have updated the manuscript with some of the suggestions we received. The most prominent changes are: 1) we rerun all the experiments in the paper with 7 random seeds [3 more seeds currently in progress!], and 2) we performed the analyses described in Section 7.3, that bring a new perspective beyond analyzing reward. The results can be found in the appendix, and we will add more discussion about them in the main text for the camera ready version.

The results with 7 random seeds are consistent with the ones we showed previously with 3 random seeds, and now a even more consistent improvement of MADDPG-*E models over RMADDPG can be appreciated. We would like to take this opportunity to reemphasize that the main goal of having models tailored specifically to our task is not just to claim an improvement over the baselines, but rather to show that even task-tailored models fail to solve this simplified scenario --as reviewer t9A3 insightfully points out. We believe that by using SymmToM as benchmark, we can better direct immediate next steps in machine theory of mind modeling research.

We are actively working on adding a specific related work section (starting with the pointers we received as feedback) while fitting our work into the page limit. We will include this section in the camera ready version.
We apologize for the delay in writing the author responses: we’ve been focused on getting results for new analyses, and improving the statistical significance of our results by running more random seeds. We hope this new version addresses the most important concerns!

Thanks again,

The authors

---

### Decision · Program_Chairs · 2022-01-20

**Decision:**

Reject

**Comment:**

The paper introduces a theory of mind benchmark.

This paper certainly improved during the discussion period. However, the paper is still incomplete. The authors are working related work (paper was not updated in this regard). The experiments still need significant work. The original submission used only 3 runs (very, very low). Although the authors bumped up the # of runs, the learning curves in the appendix feature very large and overlapping error bars, and the main table of results presented in the paper contains no measures of certainty---those a reported in a separate table in the appendix making comparison tedious. The paper has a fairly informal approach to dealing with hyper-parameters that should be discussed and improved. The reviewers pointed out (in their reviews and dialog with the authors) several ways the experiments should be extended.

The contribution of the benchmark is evaluated primarily via experiments; much work needs to be done before acceptance.